# Overcoming the threat of anti-bias interventions: Combining self-report and psychophysiological measures to capture the process of change

**Félice van Nunspeet**[1‡]*, **Esmee M. Veenstra**[1‡], **Beatriz Monteiro Graça Casquinho**[1], **Naomi Ellemers**[1], **Daan Scheepers**[1,2], **Miriam I. Wickham**[1], **Elena A. M. Bacchini**[1], **Jojanneke van der Toorn**[1,2], on behalf of The Organizational Behaviour Group[¶]

**1** Faculty of Social and Behavioural Sciences, Utrecht University, Utrecht, The Netherlands, **2** Department of Social, Economic, & Organizational Psychology, Faculty of Social Sciences, Leiden University, Leiden, The Netherlands

‡ FN and EMV are listed in alphabetical order, they contributed equally to this work.
¶ Membership of the Organizational Behaviour Group is provided in the Acknowledgments.
* f.vannunspeet@uu.nl

**Data Availability Statement:** Data is available at: https://doi.org/10.17605/osf.io/vqh57.

## Abstract

Anti-bias interventions do not always have the intended results and can even backfire. In light of research on the psychology of morality, we examined whether confronting people with evidence of their own (group's) bias causes a (psychophysiological) threat response, and how to overcome this. We focused on an intervention addressing gender bias in teacher evaluations. After assessing their own teaching evaluations, we presented student research participants ($N$ = 101; 71.3% female), in Part 1 of the intervention, with evidence of bias displayed in such teaching evaluations. This evidence either did (self-implied condition) or did not (self not-implied condition) include participants' own ostensibly biased evaluations. In Part 2 of the intervention, we asked participants to reflect on the issue of gender bias, and compared the impact of two experimental instructions. In the promotion condition, instructions referred to emphasizing how the university could try to achieve the ideal of promoting fair and just evaluations of teachers. In the prevention condition, instructions referred to highlighting the university's obligation to prevent unfair and unjust teacher evaluations. While participants verbally reflected on the intervention, during both phases (in Part 1 and Part 2) we measured their psychophysiological responses using indices of cardiovascular 'threat vs. challenge'. Then, we used self-report measures to examine participants' explicit responses to the different parts of the intervention. Results revealed that implicating the self in the occurrence of bias (Part 1) raises a psychophysiological threat response. However, emphasizing the future ideal of promoting fair evaluations of teachers (rather than the obligation of preventing biased evaluations; Part 2) resulted in a psychophysiological challenge response and increased perceived coping abilities to combat such bias. The implications of these findings are discussed.

**Funding:** This research was supported by the NWO-Spinoza prize awarded to N.E. by the Netherlands Organization for Scientific Research (NWO). https://www.nwo.nl/nwo-spinozapremie. The funder had no role in study design, data collection and analysis, decision to publish, or preparation of the manuscript.

**Competing interests:** The authors have declared that no competing interests exist.

## Introduction

Ever since Kurt Lewin [1] made a case for action research, the social sciences in general,—and social psychologists in particular—consider the potential significance of their work not only to generate theory, but also to benefit society. This resonates with the increased emphasis on scientific outreach, knowledge transfer and utilization, and public engagement urging academic researchers to enhance the impact of their findings and to connect science and society [2–4]. The widely acknowledged responsibility to generate impact manifests itself in current efforts to improve individual and societal outcomes and change behaviour by developing evidence-based interventions (see, for example [5]).

A central aim of our research group is to contribute to such evidence-based interventions—in particular interventions relating to behaviour that has moral implications, such as measures aiming to enhance diversity and inclusion in organizations. As a group we therefore devote a substantial proportion of our time and resources to developing outreach activities highlighting the implications of key research findings for practitioners. This is intended to educate non-scientific audiences about relevant insights (for instance on the persistence of (implicit) bias and how to overcome this), while enhancing our theorizing on evidence-based interventions [6, 7]. These experiences have taught us that even audiences that communicate an explicit interest in these issues often find it difficult to accept scientific evidence for the persistence of bias. This raises additional questions about what coping mechanisms people employ to deal with evidence of bias that implicates their self-views, and how this impacts their willingness to address bias. In the present study, we therefore take an experimental approach to compare different versions of the same intervention. More specifically, we measure how psychophysiological responses unfold during the intervention to capture the process of change, and we combine this with self-reports revealing how participants explicitly reflect on their experience. This allows us to specify intended and unintended consequences of the intervention, and systematically identify which aspects might be adapted to enhance the effectiveness of anti-bias interventions. This approach increases our understanding of psychological mechanisms that may either undermine or enhance the likelihood that people engage with the intervention and adapt their behaviour. Collecting such information will benefit our understanding of when and why interventions are most likely to be effective, and how to optimize the effectiveness of existing interventions.

We prepared this paper as the outcome of a joint project with many authors and collaborators. In this way, we capture the mission and exemplify the characteristic approach of our research group, in which many different skills and insights are brought together to develop science as well as practice. By using scientific methods to document the impact of outreach activities that are meant to communicate about research findings, we complete the full empirical cycle.

### Measuring the effectiveness of anti-bias interventions

Interventions to make people aware of and combat bias are becoming increasingly popular. In fact, many organizations nowadays expect members of staff in leadership and HR roles to participate in some form of '(implicit) bias training', in an attempt to foster employee diversity and inclusion. This is even obligatory in some public institutions, for instance for universities in the UK that apply for certification needed to qualify for research funding (Athena SWAN Award Scheme for Gender Equality), and for individual scholars who sit in on research evaluation committees for the European Science Foundation [8]. However, there is little evidence for the diversity gains of such awareness and training efforts, prompting researchers to characterize exchanges of 'best practices' as no more than 'best guesses' [9–11].

A feature shared by most anti-bias interventions is creating awareness of the existence and pervasiveness of (implicit) bias. For instance, real time demonstrations during the intervention are often used to convince participants that they too may suffer from biases—even when they explicitly endorse measures promoting equal opportunities (e.g., as revealed by participating in an Implicit Association Test [12], see also [13]). There might be good reasons for adopting this type of approach: Anti-bias training programs that actively invite participants to consider their own biases generally elicit more positive attitudes towards disadvantaged groups and offer more evidence of (long term) behavioural changes than more passive forms of information transfer [14, 15]. However, several other studies have documented the limited impact of such interventions [16–22]. Hence, so far there is no consistent evidence of what determines the effectiveness of anti-bias interventions in changing people's behaviours towards diversity issues [7, 17, 19, 22]. Nor do we know whether or how demonstrating that participants in the intervention also suffer from (implicit) bias contributes to these effects. Then again, many studies aiming to examine the impact of anti-bias interventions simply compare pre- and post-intervention questionnaires, rather than systematically monitoring psychological mechanisms that might enhance or impede the impact of specific aspects of the intervention [21]. For instance, no studies to date have examined real-time (physiological) responses documenting the process of change while taking part in a diversity training—the approach we took in the current study.

## The motivational state of threat

In the current research, we posit that demonstrating that participants in the intervention also suffer from bias may backfire when this raises threat responses that result in defensiveness. Instead of increasing people's motivation to change, threat responses are likely to *prevent people from engaging* with the intervention. The possibility that the self-relevance of information about the persistence of bias is threatening, can be derived from research showing that people are motivated to be moral, and to appear moral in their own eyes and the eyes of others [23–25]. Fair treatment of others is a key moral foundation [26, 27]. Accordingly, prior research has indicated that information emphasizing that people themselves, or members of their group, have treated others unfairly is a source of self-threat and guilt [28, 29]. Across many groups and contexts such information has been found to raise defensive responses. Instead of motivating people to repair harm done, or attempt change, they blame the victim or find ways to justify past injustices [30–35]. Likewise, people have been found to respond in a defensive manner to feedback indicating they themselves are biased [36–38]. *The first aim of the current research is therefore to test our hypothesis that when people are confronted with evidence of their own bias—compared to when evidence of existing bias is not implying the self—they will display threat and defensive responses.*

## The motivational state of challenge

Improving our understanding of what can be done to overcome such threat makes it possible to enhance the effectiveness of interventions that aim to reduce (the effects of) social bias. Interestingly, prior work suggests that defensive responses to feedback indicating that people themselves are biased are not inevitable. In general, constructive responses to information about implicit bias can be enhanced when people are offered a vision for improvement instead of focusing only on problems that should be avoided [34, 39–41]. Here, we extend prior research examining how different conditions may encourage individuals to overcome (implicit) bias, by reducing the defensive responses that characterize the experience of moral threat. Specifically, we examine specific ways of communicating about the persistence of

implicit bias that are likely to make it easier for people to cope with and respond to this (threatening) information.

We build on regulatory focus theory [42], distinguishing a focus on *promoting* positive outcomes and ideals from the focus on *preventing* negative outcomes that violate important obligations [43]. Previous research suggests that encouraging a focus on promoting positive outcomes rather than preventing negative outcomes can help avoid defensive responses. This also seems to be the case when evidence of pervasive bias is presented. Shifting people's focus on the pursuit of ideals (promotion-focus) as opposed to obligations (prevention-focus) can help to combat defensive reactions and make people more receptive to (potentially) threatening information [39, 40]. Specifically, results from prior research examining how student participants responded to statistics documenting labour discrimination demonstrated that focusing on ideals (i.e., a promotion-focus; as opposed to obligations, indicating a prevention-focus) elicits responses indicating relatively more challenge and less threat [39, 40]. These results suggest that focusing on ideals to be achieved (inducing a promotion focus) as opposed to obligations to be met (prompting a prevention focus), can diminish threat responses and improve people's ability to engage with attempted change. *Our second aim with the current research is therefore to test whether, and further examine why, promoting the **ideal** of fairness and equal treatment—rather than the **obligation** to prevent unfairness and unequal (i.e., biased) treatment—elicits a challenge response.*

Furthermore, past research has shown that offering people feedback that highlights potential future opportunities, for instance to reveal their moral character or behavioural intentions instead of past (immoral) events, raises their perceived ability to cope with the situation and allows them to engage with opportunities for change [44]. In a similar vein, emphasizing that past failures may indicate a lack of competence instead of moral inadequacy makes people more confident that they will be able to improve their behaviour and social image [45, 46]. *Our third aim with the current research is therefore to explore whether and how the effect of the promotion (vs. prevention) focus is associated with people's perceived coping ability.*

## The biopsychosocial model of challenge and threat

The biopsychosocial model of challenge and threat (BPS-CT) specifies cardiovascular (CV) markers of the motivational states of *threat* and *challenge* during so-called *motivated performance situations* (e.g., athletic performance, negotiating, doing a math test, giving a speech [47–52]). In addition, the BPS-CT characterizes validated CV response-profiles that mark challenge rather than threat. In the context of the BPS-CT, four CV measures are typically used: Heart rate (HR), Pre-Ejection Period (PEP), Cardiac Output (CO) and Total Peripheral Resistance (TPR). A defining aspect of motivated performance is a certain level of task engagement which is at the physiological level marked by increased HR and decreased PEP [48, 51, 53]. That is, the heart starts pumping faster, and with more force. Given task-engagement, challenge and threat can in turn be distinguished based on CO and TPR. In terms of reactivity compared to baseline levels (so-called absolute patterns of challenge and threat [48, 54]), challenge is marked by increased CO and decreased TPR, which facilitates the efficient mobilization and transportation of energy during motivated performance. Threat, by contrast, is marked by increased TPR and stable CO, which is a less efficient CV response profile during motivated performance [48].

In addition to these absolute patterns of challenge and threat, in more recent formulations of the BPS-CT more stress is placed on *relative* differences in challenge and threat between experimental conditions. Here, challenge is indicated by relatively high CO and low TPR, and threat by relatively high TPR and low CO. As a further elaboration of these relative differences

in challenge and threat motivational states, CO and TPR can also be combined in a single threat-challenge index (TCI [55–60]).

## The current research

In the current research, we combine insights from psychological theory and experimental research to examine how different versions of an anti-bias intervention impact on participants' thoughts, feelings and experiences. Because of our aim to examine a real-life intervention, student research participants were confronted with existing evidence on a form of bias relevant to them: gender bias in student evaluations of teachers. This bias indicates that students systematically give lower teaching evaluations to female as opposed to male teachers—irrespective of objective differences in teaching effectiveness [61, 62]. Except for selecting a theme that would be relevant to the target group, the procedure we followed is very similar to how we typically present evidence of bias displayed by one's own group in anti-bias interventions we conduct in organizations (e.g., during public outreach activities).

Prior to taking part in the intervention (which was embedded in an interactive webinar in the lab about measuring teaching effectiveness), student participants were invited to complete an online questionnaire and evaluate teachers of recently attended courses. During the webinar, it was first explained that student evaluations of teachers, which are regarded as one of the most important indicators of teaching effectiveness, impact on the career progression of teachers. Then the question was raised whether these student evaluations indeed reflect teaching effectiveness, and research into the teacher evaluation bias was introduced. In the remainder of the webinar, participants were educated about the nature, pervasiveness, and impact of this form of gender bias, based on the growing body of evidence that evaluations of teachers tend to reveal gender bias instead of accurately reflecting teaching effectiveness [61, 63, 64].

After this general introduction, in Part 1 of the intervention, participants in the self-implied condition were presented with results allegedly indicating that their own evaluations of teachers were not immune to gender bias. Specifically, they received feedback specifying that they and their fellow students had shown a gender bias in the evaluation of teachers to the disadvantage of female teachers, ostensibly based on the online questionnaire they had completed prior to the experiment. We compared this to a condition with no such self-implication, in which participants only received the general information on the likelihood that such biases emerge, in the webinar. Subsequently, in Part 2 of the intervention, we randomly exposed research participants to one of two versions of our anti-bias intervention, with the promotion version emphasizing the ideal of promoting fair and just evaluations of teachers, and the prevention version highlighting the obligation of preventing biased evaluations of teachers.

In both parts of the intervention, we invited participants to verbally reflect on the existence of the bias in teaching evaluations (Speech 1) and the promotion of fair (vs. the prevention of unfair/biased—Speech 2) treatment of both male and female teachers, while measuring their cardiovascular responses. Subsequently, we asked participants to complete several self-report questionnaires assessing their affective responses to the experimental parts of the intervention, and their behavioural intentions after having completed the intervention. By examining their underlying motivational states (i.e., real-time cardiovascular indices of threat vs. challenge), as well as more traditional self-reported reactions, we were able to assess the added value of more implicit indicators of participants' responses to their explicit evaluations of various aspects of the intervention.

In sum, this study was designed to test the following hypotheses:

1. Being confronted with self-implied (vs. self not-implied) evidence of the existence of gender bias in teacher evaluations, will elicit a threat response among students (H1).

2. Inviting students to promote the fair and just (instead of requesting them to prevent biased) treatment of both male and female teachers will elicit a challenge response among students (H2).

3. Occurrence of the challenge response elicited by the promotion focus will be mediated by students' perceived coping ability to combat the teacher evaluation bias (H3).

In addition to testing these hypotheses, we explored how the (more implicit motivational) psychophysiological responses relate to (more explicit) self-reported reactions to the intervention–which are typically used in standard evaluations of such interventions.

## Method

### Ethics statement

This entire research project, including the hypotheses and the exploratory analyses described above, was approved by the local Ethics Committee of the Faculty of Social and Behavioural Sciences of the university. Research participants were recruited and data was collected between July 14—September 16, 2018. All participants gave their written informed consent.

### Participants and design

Based on prior research [40] in which cardiovascular responses were measured and—comparable to our promotion vs. prevention manipulation—moral ideals vs. obligations were emphasized, a power analysis was conducted. The smallest effect size reported of a cardiovascular effect comparing a promotion and prevention condition (i.e., an emphasis on moral ideals vs. moral obligations [40]) was $\eta p^2 = 0.12$, Cohen's $f = 0.37$. Based on this effect size and our aim to compare the effects of two different manipulations (i.e., self-implied vs. self not-implied evidence, and a promotion vs. prevention frame) on cardiovascular markers for threat and challenge in a one-way ANOVA, a power analysis using G*Power indicated that we needed to include around 84 participants in total to obtain a power of 80%. We recruited additional participants to account for unintended data loss at the start of the data collection, due to technical errors, and participants not completing the experiment or not strictly following the procedure. Specifically, 108 participants took part, who were all students at a Dutch university and who were financially compensated for their time or received course credit for participation. Seven participants were excluded for not completing the experiment or not strictly following the procedure. For 18 participants, the monitoring of their cardiovascular measures was *partially* affected due to technical errors in the initial phase of the data collection. We therefore decided for each analysis, to include only the participants of whom enough good-quality data remained, resulting in small differences in sample sizes across analyses.

Of the remaining 101 participants ($M_{age} = 23.40$ years, $SD = 3.00$, age range = 18–39 years), 71.3% indicated being female, 27.7% male and 1% 'other'. Most participants (53.5%) were Master students, 43.6% of participants were Bachelor students (5% in year 1, 8.9% in year 2, and 29.70% in year 3/4), and 3% of participants were working towards another type of degree. The native language of most participants was Dutch (57.4%), English for 9.9%, and a different native language for 32.7%. Participants were randomly assigned to one of four conditions in a 2 (Self-implied evidence: Yes vs. No) × 2 (Frame: Promotion vs. Prevention) between-participants design.

## Procedure

A complete overview of all phases of the experimental session is visualized in Fig 1. Participants were recruited for 'a study on the effectiveness of teaching evaluations' and agreed to take part in a psychophysiological lab study. The study began, on average, five days prior to the lab session ($M_{days}$ = 4.72, $SD$ = 4.46) when participants completed an online survey in which they were asked to evaluate four different teachers of courses they had recently attended (details of this assessment, as well as the results, are reported in S1 Fig in S1 Appendix).

During the lab session, participants were informed that they would be shown a webinar on measuring teaching effectiveness—presented by a (white, female) professor in social psychology at the participants' university (who is a member of our research team and an expert on diversity and inclusion), and that their cardiovascular responses would be measured. One of seven different experimenters (all of whom were White and female) applied the sensors for the cardiovascular measurements (see below). Importantly, none of the researchers (i.e., the experimenters and the university professor who presented in the webinar), was included in participants' teaching evaluations as these evaluations concerned courses outside of the researchers' teaching responsibilities. Independence between the researchers and the research participants was thus assured. Once participants were connected, they were further informed about the different parts of the experimental session: It was explained that they would be asked twice to verbally reflect on the message that was presented to them in the webinar, and that they would be asked to complete several questionnaires at the end of the session (after the entire webinar). Participants then watched a 5-minute relaxing video during which we measured their baseline heart rate, which was followed by the start of the webinar.

In the first part of the webinar, the professor explained the relevance for and impact of students' teaching evaluations on teachers' career progression. She also presented several examples of published research evidence suggesting that such evaluations are often biased against female teachers. Participants in the self not-implied condition did not receive any additional information. Participants in the self-implied condition were also presented with a brief summary indicating they and their fellow students had shown a gender bias in their teacher evaluations–allegedly based on the online questionnaire they had filled out before they had come to the lab. This feedback was not presented within the webinar (i.e., by the professor), but rather provided as a separate (additional) section of the experiment. After this first part of the intervention, participants were asked to give their first speech (Speech 1), during which

| (0) | (1) | (2) | (3) | (4) | (5) |
|---|---|---|---|---|---|
| Online Questionnaire | Webinar 1 | Speech 1 | Webinar 2 | Speech 2 | Survey |
| *Evaluation of own teachers* | *Information about the existence of gender bias in teacher evaluations* | *Where does this bias come from?* | *How to: achieve the ideal of promoting fair / meet the obligation to prevent unfair evaluations* | *Please list your ideas* | *Self-reports* |

**Fig 1. The (parts of the) experimental intervention.**

cardiovascular responses were registered. They were then shown the second part of the webinar, which focused on the implications of this gender bias for the university and the university's reaction to the research findings. Depending on their random assignment to experimental condition, participants either saw a version which was framed in terms of a promotion focus or a version framed in terms of a prevention focus. Then, participants were asked to verbalize their thoughts about this second part of the intervention (Speech 2), after which they completed the self-report questionnaires. Finally, they were fully debriefed, compensated, and thanked for their participation. In the debriefing, it was explicitly emphasized that bogus evidence of bias had been presented and that this did not reflect participants' actual answers given in their teacher evaluations.

### Instruments and measures

**Self-implication manipulation.** Participants were randomly assigned to either the self-implied or the self not-implied condition. In the self-implied condition, participants were presented with bogus results, ostensibly based on responses to the teacher evaluation survey they had participated in, demonstrating that they and their fellow students showed a bias in the evaluation of teachers to the disadvantage of female teachers. In the self not-implied condition, these results were not shown. After this, participants were asked to give a speech (Speech 1) in which they verbally reflected on the question: "Where do you think the bias in the evaluation of teachers comes from?" Participants had one minute to prepare, and then two minutes to subsequently deliver their speech.

**Framing manipulation.** Participants watched one of two versions of the second part of the webinar depending on whether they had been randomly assigned to the promotion or prevention condition. In this part of the webinar, the professor talked about the reaction of the students' university to the research on the teacher evaluation bias and how the university and the student council wanted to respond to it. In the promotion condition, it was emphasized that the research findings offered *opportunities* to *improve* the situation and helped the university to *achieve* its *ideal* of promoting fair and just evaluations of teachers. Conversely, in the prevention condition, the professor stressed that the research findings pointed out a situation that the university wanted to avoid in order to meet its *obligation* of *preventing* unfair and unjust evaluations of teachers. This promotion vs. prevention perspective was also reflected in the reflection assignment participants received for the second speech (Speech 2). In the promotion condition they were asked: "What ideas can you come up with to promote the fair evaluation of teachers? Please generate ideas to achieve the ideal of fair and just evaluations of teachers". In the prevention condition they were asked: "What measures can you come up with to prevent the unfair evaluation of teachers? Please list measures to meet the obligation to prevent biased evaluations of teachers." Again, participants received one minute to prepare and two minutes to give the speech.

**Cardiovascular measures.** Throughout the experiment we continuously measured CV-markers using a Biopac MP160 system [65]. Specifically, we used the NICO100C module to measure impedance-cardiography (ICG), the ECG100 module to measure electrocardiography measures (ECG), and the NIBP100D module to measure blood pressure (BP). We followed the procedures previously reported in [66]. Physiological data were stored using *Acqknowledge* software [65] and scored using the physiodata toolbox [67]. Specifically, ECG, ICG, and BP signals were first visually inspected for artifacts; complexes that were non-scorable or otherwise of low quality were rejected. The r-peaks of the ECG were automatically scored which yielded a measure of heart rate (HR). The BP signal was also automatically scored which yielded a measure of mean arterial pressure (MAP). The ECG and ICG were then ensemble-

averaged for the three epochs we were interested in: The last minute of the baseline, and one minute of the first and the second speech task (seconds 10 through 70), respectively. The Q, B, and X points were then manually scored in the ensemble-averaged signals; the Q-point was scored as 'r-onset' [68] and the B- and X-point were scored following [69]. This yielded the pre-ejection period (PEP, which is a measure of ventricular contractility) and cardiac output (CO). Finally, in SPSS, TPR was calculated from CO and MAP using the following formula: $TPR = (MAP/CO) \times 80$ [70].

Also in line with standard practice, individual reactivity scores were created for the four measures by subtracting the baseline scores from each of the six task scores. Univariate outliers (defined as 3.3SD above/below the mean) were assigned a value of 1% higher/lower than the adjacent non-extreme value [55, 71, 72]. Finally, we calculated combined Threat-Challenge Indices (TCI) by calculating Z-scores of CO and TPR reactivity, then multiplying TPR with -1 and summing the result with the CO Z-score [55, 56, 60]. Higher scores on the resulting indices—which maximize the reliability of the cardiovascular measures [60]— indicate a greater challenge motivational state, whereas lower scores indicate a greater threat motivational state.

**Self-report measures.** At the end of the webinar (i.e., following participants' second speech), we used self-report questionnaires to assess participants' explicit responses. We did this to test the subjective equivalence of experimental conditions, and to be able to rule out alternative explanations, as well as to explore relations between these standard self-reported evaluations and more implicit cardiovascular indicators of participants' responses to the intervention. Here we asked them to indicate how they had experienced the first and/or second part of the intervention, and to register their attitudes towards the general topic of the intervention (i.e., gender bias). All items were rated on a 7-point Likert scale (1 = strongly disagree–7 = strongly agree), unless otherwise indicated.

*Manipulation checks.* To examine whether the self-implication manipulation evoked explicit defensive reactions, participants in the self-implied condition indicated their agreement with statements adapted from the work on defensive responses to feedback regarding individual's implicit biases [73, 74]. The total defensiveness scale of nine items had a relatively low reliability (α = .65) and was hence subjected to a Principal Components Analysis with Varimax rotation, which yielded three orthogonal components. The first factor represents the degree to which the participants rejected that *the feedback represented their deliberate*, *true preferences* (α = .85), and included three items such as: "The results from the online questionnaire that I and my fellow students filled out:. . .do not reflect anything about my true thoughts or feelings". The second factor represents the degree to which the participants rejected *the feedback being corrupt* (α = .66) and included four items such as: "The results:. . .are misleading". The third factor represents the degree to which the participants rejected that *the feedback reflected their unconscious*, *automatic preferences* (two items, $r$ = .80), with an example item being: "The results:. . .reflect something about my automatic thoughts or feelings" (reverse coded). The three-factor solution explained 68.8% of the variance.

To examine the effect of the framing manipulation, six items were included to assess whether participants were focused on approaching opportunities (promotion focus) or avoiding risks (prevention focus) during Speech 2. Five items based on the regulatory strategy scale [75] were used, each with a promotion strategy at one end and a prevention strategy at the other end of the scale. For example, participants responded to the following contrasting tendency: "During the second speech, I was focused on:. . .taking risks" versus "acting cautiously". Participants had to answer using a 7-point scale, in which a rating of 4 would indicate that both prevention and promotion strategies fit to the same extent and lower values would indicate a better fit of promotion strategies. We added the item: ". . .taking opportunities" versus

"taking responsibility" (six items, $\alpha$ = .34). To improve the internal consistency, we decided to exclude one item (acting thoroughly/superficially) from the total scale ($\alpha$ = .64).

*Perceived coping abilities*. Related to our third hypothesis, we examined participants' ability to cope with our requests to verbally reflect on the two issues raised in the webinar. We did this using two items [adapted from 76] assessing perceived demands and resources in a particular situation. In addition, one item was added to incorporate the perceived difficulty of our request. That is, regarding Speech 1 ($\alpha$ = .69), participants responded to the items: "In preparation of and during the first speech, . . . it was stressful to come up with an answer to the question where the bias in the evaluation of teachers comes from" (measuring *demands*, reverse coded) ". . .I felt I was able to reflect on the bias in the evaluation of teachers" (measuring *resources*), and ". . .it was difficult to come up with a response to the question where bias in the evaluation of teachers comes from" (measuring *difficulty*, reverse coded).

Similarly, regarding Speech 2 ($\alpha$ = .76), participants responded to the items: "In preparation of and during the second speech, . . . it was stressful to come up *with ideas to promote the fair evaluation of teachers / measures to prevent the unfair evaluation of teachers*" (measuring *demands*, reverse coded) ". . . I was able to *generate ideas to achieve the ideal of fair and just evaluations of teachers / list measures to meet the obligation to prevent biased evaluations of teachers*" (measuring *resources*), and ". . .it was difficult to come up with *ideas to promote the fair evaluation of teachers / measures to prevent the unfair evaluation of teachers*" (measuring *difficulty*, reverse coded).

*Additional self-report measures*. In addition to the manipulation checks and the role of coping, we also explored effects of the experimental manipulations on participants' intention to act and motivation to change.

*Willingness to take action*: Participants were asked whether they wanted to be involved in a student panel to talk about the future of student evaluations of teachers. They were given the following seven options for concrete choices they might make (percentages are given in parentheses): "*Yes, I would like to engage in a conversation with the student council to find out how I can contribute*" (15.3*%*), *"Yes, I would like to attend one of the information sessions to hear what the student council expects from students"* (20.4%), "*Yes, I would like to stay informed on the composition of this panel by subscribing to the monthly newsletter of the student council*" (19.4%), *"No, I would like to forget about the bias in the evaluation of teachers and its consequences"* (1%), *"No, I rather not get mixed up in discussions about the bias in the evaluation of teachers to the disadvantage of female teachers"* (5.1%), "No, I cannot devote my time to this, as I have other important things to do" (33.7%), and *"Other, namely. . ."* (5.1%).

Preliminary analyses using post-hoc z-scores were used to examine the specific differences in response patterns to the provided options [77]. This revealed no significant differences between the first two answers ($p$'s > 0.05), which could both be considered active ways of contributing to the panel, and which were therefore combined into one category—indicating *active involvement*. Results neither revealed significant differences between the fourth and fifth answers ($p$'s > 0.05), which could both be considered active ways of not contributing to the panel, and which were therefore also combined into one category—indicating *active distancing*. This thus resulted in the following four categories: 'active involvement', 'passive involvement', 'active distancing', and 'passive distancing'. Further analyses are discussed in the results section.

*Future intentions to regulate bias*: Three items, inspired by [38], assessed participants' future intentions to monitor and control their possible teacher evaluation bias. Example items are: *"I am concerned I will exhibit gender bias in the future"*, and *"After what I learned today, I will be more on guard for gender biased behaviour"* ($\alpha$ = .62).

*Belief in present gender discrimination*: Six items from the Contemporary Gender Discrimination Scale [78] were included to examine participants' belief in present gender

discrimination. For example, *"Although it is more subtle than it used to be, women still experience discrimination."* and *"Women still need to work harder than men to achieve the same things.".* Because three of these items were reverse-coded, we first conducted a factor analysis to determine the scale's dimensionality. We used principal axis factoring with a Principal Components Analysis with Varimax rotation. Results indicated that all items loaded on one factor (48.19% variance explained). The scale had sufficient reliability ($\alpha$ = .76).

Besides the measures described above, participants were also presented with self-report measures regarding the perceived stability of bias, threat to one's social identity, positive and negative reactions to teacher evaluation bias, attention checks and experiences in the experiment. Furthermore, we analysed the speeches that participants had to produce in terms of content. Because these measures go beyond the scope of our main manuscript, but do not alter (the interpretation of) our findings, details are included in S1–S3 Appendices.

## Results

### Manipulation checks

The self-report measures collected at the end of the intervention asked participants in the self-implied condition to reflect back on the feedback they had received—presenting (ostensible) evidence that they themselves and their group members had shown a gender bias in their teacher evaluations. Results indicated that participants did not believe that the feedback represented their true values or reflected their true thoughts or feelings (i.e., $M$ significantly higher than scale midpoint, $M_{\text{diff}}$ = 0.55, $t$[49] = 2.35, $p$ = .023, 95% CI [0.079, 1.027], Hedges's $g_{\text{s}}$ = 0.654). However, participants did not invalidate the feedback (i.e., $M$ significantly lower than scale midpoint, $M_{\text{diff}}$ = -0.53, $t$[49] = -3.31, $p$ = .002, 95% CI [-0.852, -0.208], Hedges's $g_{\text{s}}$ = -0.922). This indicates that they thought the feedback about their own (group's) bias was based on fact and not misleading or exaggerated. Moreover, interestingly, participants did not fully reject the feedback (i.e., $M$ somewhat lower than scale midpoint, $M_{\text{diff}}$ = = -.39, $t$[49] = -1.89, $p$ = .064, 95% CI [-0.804, -0.024], Hedges's $g_{\text{s}}$ = -0.526, suggesting that they saw the results as reflecting something about their automatic thoughts or feelings. These findings were not moderated by the framing manipulation (all $p$'s > 687). Together, these results reveal that participants believed the feedback to be factual, but indicative of an implicit, rather than explicit, gender bias.

We also examined whether participants in the promotion and prevention conditions deliberately intended to approach opportunities or to avoid risks while they came up with ways to address gender bias in the future—during their second speech. Results of this self-report measure indicated that participants in the promotion condition did not think they had been particularly focused on approaching opportunities ($M$ = 4.00, $SD$ = 0.93), $t$(52) = 0, $p$ = 1.00), nor did participants in the prevention condition indicate being particularly focused on avoiding risks ($M$ = 3.78, $SD$ = 0.84), $t$(47) = -1.85, $p$ = .071). This suggests that participants' explicit recall of their deliberate intentions during the second speech did not differ depending on experimental condition, $F$(1,97) = 1.59, $p$ = .21. We now continue by testing our hypotheses about the psychophysiological responses during each of the phases of the intervention.

### Hypothesis testing

Before testing our hypotheses regarding participants' cardiovascular responses, we first verified that any differences found between conditions can be attributed to the experimental manipulations in the intervention, rather than to participants' task engagement. We therefore examined participants' *behavioural task engagement* by inspecting the number of words spoken and the number of hesitance words they used during both speeches, and their *cardiovascular signs of*

*task engagement* during both speeches, by inspecting their CV-reactivity in indices of HR and PEP compared to baseline. Results regarding the cardiovascular task engagement indicated that participants were more engaged during both speeches compared to the baseline period (for HR: $t$'s $\geq$ 11.07, $p$'s $\leq$ .001; for PEP: $t$'s $\geq$ -6.1, $p$'s $\leq$ .001). Moreover, regardless of the condition participants were in, they showed similar levels of behavioural (all $t$'s $\leq$ 1.72, $p$'s $\geq$ .09) and cardiovascular task engagement (all $t$'s $\leq$ 1.94, $p$'s $\geq$ .055) during the speeches.

To test our hypothesis regarding the psychophysiological measures, we first calculated mean levels of HR, PEP, CO and TPR for the last minute of the baseline period, as well as the first minute of the two speech tasks, respectively, and checked for between-condition differences. We found a significant interaction between self-implication and framing on baseline HR, $F(1, 95) = 4.82$, $p = 0.031$. In line with standard practice, we therefore included baseline HR as a covariate in the confirmatory analyses on the cardiovascular data reported below. However, it should be noted that the results are similar when this covariate is not included. No other main or interaction effects were found, $F$'s $\leq$ 1.84, $p$'s $\geq$ .179.

**Effects of self (not) implying evidence (Speech 1).** Our first hypothesis stated that being confronted with self-implied (vs. self not-implied) evidence of the existence of gender bias in teaching evaluations, should elicit a threat response. To test this prediction, we examined the *relative* patterns of challenge and threat by performing an ANOVA on the separate cardiovascular indicators TPR and CO, as well as on the combined TCI as an overall indicator of relative challenge vs threat—all measured during the first speech—and included our experimental manipulation of self-implied vs. self not-implied evidence of bias as the independent variable. Results showed no between-condition differences in TPR, CO, and TCI ($F$'s $\leq$ 2.41, $p$'s $\geq$ .12).

However, in line with common practice [e.g., 54], we also examined *absolute* patterns of cardiovascular (CV) reactivity as changes in TPR and CO during the first speech task compared to baseline levels. Consistent with the typical pattern related to threat, results revealed a significant increase in TPR in the self-implied condition, $t(42) = 2.42$, $p = .020$, while TPR was stable in the self not-implied condition, $t(39) = 0.28$, $p = .784$. Additionally, although CO was increased in both the self-implied ($t[42] = 2.70$, $p = .010$) and the self not-implied condition compared to baseline ($t[43] = 4.69$, $p < .001$)—whereas the typical pattern related to threat is associated with CO remaining stable—, the increase in the self-implied condition was smaller ($M_{diff} = 0.25$, $SD = 0.61$) than the increase in the self not-implied condition ($M_{diff} = 0.45$, $SD = 0.64$). These results thus indicate that, in line with Hypothesis 1, reflecting on evidence of bias that implies the (group-) self yields CV reactivity indicative of threat (i.e., higher TPR and relatively lower CO). The mean levels of CO, TPR, and TCI during Speech 1 (relative to baseline) are displayed in Table 1.

**Effects of a promotion vs. Prevention frame (Speech 2).** Our second hypothesis was that a promotion (vs. a prevention) frame, for how to address the equal treatment of teachers, should elicit a challenge response. To test this prediction, we examined the *relative* patterns of challenge and threat by performing ANOVA's on the separate cardiovascular indicators TPR and CO, as well as on the combined TCI as an overall indicator of relative challenge vs threat —measured during the second speech. Since the second speech was performed after the whole intervention had been completed, we included both experimental manipulations (promotion vs. prevention frame, as well as self-implied vs. self not-implied evidence of bias) as independent variables. Results showed no effects of the framing, nor the self-implication manipulation, on TPR during the second speech, $F$s $< 1.80$, $p$s $> .182$. Regarding CO, we found no main effect of self-implication, $F(1, 82) = 0.86$, $p = .357$, nor an interaction effect between self-implication and framing, $F(1, 82) = 0.89$, $p = .349$. However, in line with our hypothesis, there was a main effect of promotion vs prevention frame on CO, $F(1, 82) = 4.39$, $p = .039$, indicating higher CO in the promotion ($M = 0.16$, $SD = 0.51$) than in the prevention condition ($M =$

**Table 1. Cardiovascular reactivity as a function of self (not) implying evidence during Speech 1.**

|  | Self-implied | | Self not-implied | |
|---|---|---|---|---|
|  | $M_{diff}$ | SD | $M_{diff}$ | SD |
| CO | 0.25* | 0.61 | 0.45*** | 0.64 |
| TPR | 177.24* | 480.22 | 24.96 | 571.21 |
| TCI | -0.27 | 1.62 | 0.29 | 1.76 |

*Note.* Condition means were tested against zero to determine significant increases or decreases from the baseline for CO = Cardiac Output, and TPR = Total Peripheral Resistance. TCI = Threat-Challenge Index, reflected in Z-scores, with lower scores indicating relatively more threat, higher scores more challenge.

* $p < .05$.

** $p < .01$.

*** $p < .001$.

-0.04, $SD = 0.40$). Even more interesting, results showed a main effect for frame on the combined TCI, indicating that participants in the promotion condition were relatively more challenged ($M = 0.35$, $SD = 1.64$), than participants in the prevention condition, who were relatively more threatened ($M = -0.37$, $SD = 1.72$), $F(1, 78) = 4.21$, $p = .043$. Neither the main effect of the self-implication manipulation, nor the interaction effect between self-implication and framing were significant, $F$'s $< 1.49$, $p$'s $> .226$. Mean difference levels of CO, TPR, and TCI reactivity during the second speech, as a function of self-implication and framing, are displayed in Table 2. The pattern for the combined TCI is displayed in Fig 2.

**Perceived coping abilities.** Finally, we tested our third hypothesis, that occurrence of the challenge response when reflecting on the promotion of the fair and just (instead of the prevention of unfair or biased) treatment of both male and female teachers, would be mediated by students' perceived coping abilities. To do this, we first examined whether there were any

**Table 2. Cardiovascular reactivity as a function of self-implication and frame during Speech 2.**

|  | Promotion | | Prevention | | Total | |
|---|---|---|---|---|---|---|
|  | $M_{diff}$ | SD | $M_{diff}$ | SD | $M_{diff}$ | SD |
| **CO** |  |  |  |  |  |  |
| Self-implied | 0.09 | 0.58 | -0.05 | 0.37 | 0.02 | 0.49 |
| Self not-implied | 0.23* | 0.44 | -0.02 | 0.43 | 0.10 | 0.44 |
| Total | 0.16* | 0.51 | -0.04 | 0.40 | 0.06 | 0.47 |
| **TPR** |  |  |  |  |  |  |
| Self-implied | 218.70** | 352.36 | 495.69** | 733.01 | 350.90 | 577.08 |
| Self not-implied | 206.89 | 571.29 | 207.08* | 414.92 | 206.98** | 494.71 |
| Total | 213.21** | 461.21 | 358.60*** | 613.44 | 283.28 | 541.54 |
| **TCI** |  |  |  |  |  |  |
| Self-implied | 0.20 | 1.60 | -0.62 | 1.90 | -0.19 | 1.78 |
| Self not-implied | 0.52 | 1.71 | -0.11 | 1.51 | 0.22 | 1.63 |
| Total | 0.35 | 1.64 | -0.37 | 1.72 | 0.00 | 1.71 |

*Note.* All condition means were tested against zero to determine significant increases or decreases from the baseline; CO = Cardiac Output, TPR = Total Peripheral Resistance, TCI = Threat-Challenge Indices, these reflect Z-scores, with lower scores indicating relatively more threat, higher scores more challenge.

* $p < .05$.

** $p < .01$.

*** $p < .001$.

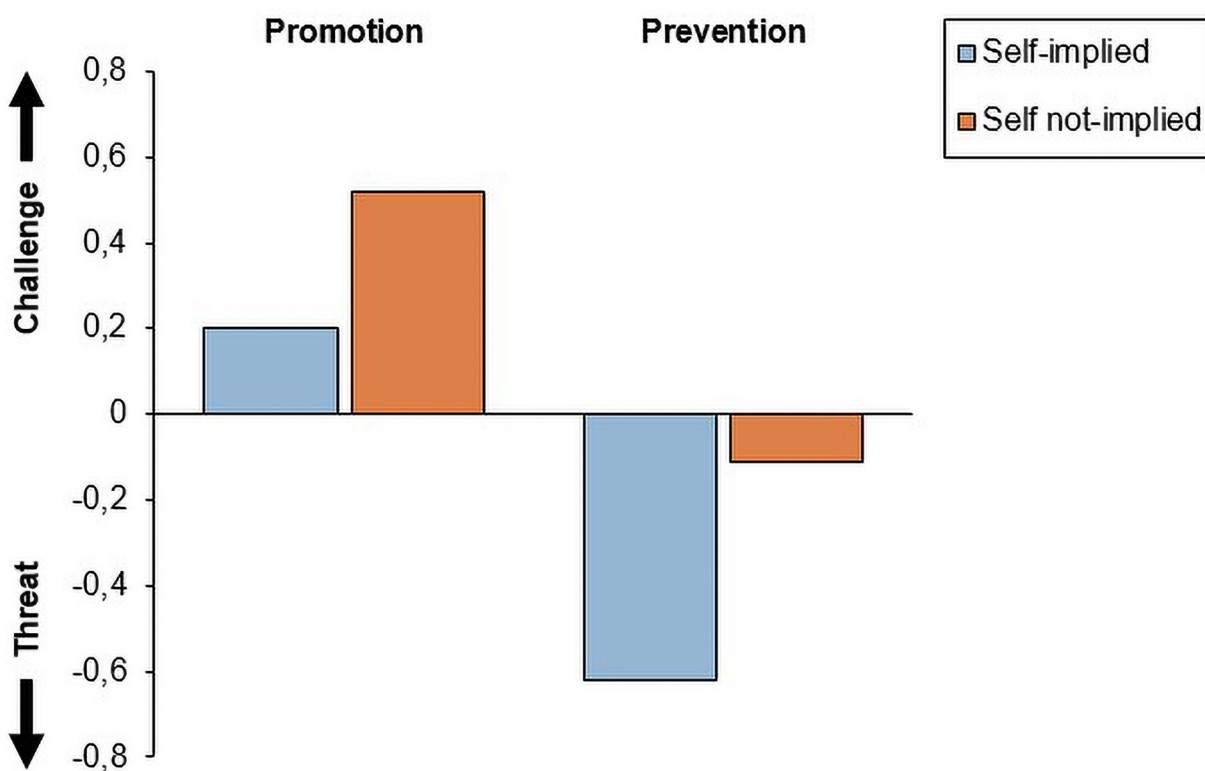

**Fig 2. Threat-Challenge Index (TCI) as a function of evidence (self-implied vs. Self not-implied) and framing (promotion vs. Prevention) during Speech 2.**

differences between conditions on participants' self-reported coping abilities regarding both the first and the second speeches. Results indicated there were none (all $F$'s $\leq$ 2.66, all $p$'s $\geq$ .11). Due to the absence of a main effect of framing on participants perceived coping abilities in their second speech, we could not test for mediation.

Exploratively, we then examined whether combined TCI during the second speech (for which we found the main effect of frame as reported above) correlated with perceived coping abilities. Results showed no significant overall correlation between TCI during the second speech and perceived coping ability, although it was in the expected (positive) direction $r(83)$ = .18, $p$ = .11. In line with our hypothesis, an additional test of the within-cell correlations showed a positive correlation between coping ability and the TCI in the promotion condition, $r(43)$ = .33, $p$ = .029, while in the prevention condition this relation was not significant, $r(40)$ = .12, $p$ = .468. Thus, enhanced perceived coping abilities during the speech in the promotion condition, were related to a stronger challenge response, while there was no reliable relation between these two variables in the prevention condition.

### Additional self-report measures

**Willingness to take action.** We explored whether participants would be more willing to contribute to a student panel to deal with the issue of gender bias in teacher evaluations, depending on the confrontation with (self-implying) evidence of bias or the promotion versus prevention frame. To this end, we conducted a chi-square test with the self-implication and framing manipulation as the independent variables and the four answer-categories (i.e., active or passive involvement in the student panel, and active or passive distancing from contributing

to the student panel) as the dependent variable. The analysis only yielded a significant result for the self-implication manipulation ($X^2$ = 10.64, $p$ = .014, $phi$ = 0.34). Post-hoc analyses using z-scores [77] indicated that participants who had been confronted with self-implying evidence of bias tended more towards active involvement in how to deal with the issue of gender bias in teacher evaluations (by taking part in the panel) or to passively distance themselves from the issue (by indicating having other things to do). However, participants who were not confronted with evidence of their own (group's) bias tended more towards passive involvement in how to deal with the issue (by subscribing to a newsletter) or to actively distance themselves from the issue (by indicating to not wanting to get mixed-up in discussions like these (all $p$'s < .05). There were no significant differences between conditions in how often participants said yes or no ($p$'s > 0.05), nor did we find any effects for the promotion vs. prevention framing ($X^2$ = 0.57, $p$ = .903, phi = 0.08).

**Future intentions to regulate bias.**    We also explored whether participants' deliberate intentions to regulate (their own) bias were different depending on the confrontation with (self-implying) evidence of bias or the promotion versus prevention frame. A 2 (Self-implied: Yes vs. No) × 2 (Frame: Promotion vs. Prevention) ANOVA revealed no significant main effect of self-implication, $F(1,97)$ = 0.24, $p$ = .624, $\omega p^2$ = -0.007, framing, $F(1,97)$ = 0.11, $p$ = .740, $\omega p^2$ = -0.009, or interaction, $F(1,97)$ = 1.41, $p$ = .238, $\omega p^2$ = 0.004 on self-stated intentions. This indicates that participants' stated concerns about (potentially) exhibiting gender bias or monitoring their own (potentially biased) behaviour were not affected by the experimental manipulations in our intervention.

**Belief in present gender discrimination.**    To examine whether participants in the promotion condition reported greater belief in present gender discrimination than participants in the prevention condition and whether this differed across the self-implication conditions, we conducted a two-way ANOVA. Results revealed no main effect of the self-implied manipulation, $F(1, 97)$ = 3.06, $p$ = .083, $\omega p^2$ = 0.02, nor an interaction effect, $F(1, 97)$ = .23, $p$ = .630, $\omega p^2$ = -0.007. Interestingly however, there was a main effect of the framing manipulation, $F(1, 97)$ = 3.27, $p$ = .074, $\omega p^2$ = 0.022, indicating that participants in the promotion condition reported greater belief in present gender discrimination ($M$ = 5.49, $SD$ = 0.86) than participants in the prevention condition ($M$ = 5.13, $SD$ = 1.01). These findings thus reveal that a promotion focus on how to address gender bias in teacher evaluations may help to open one's eyes to the persistence of the problem of social bias.

## Discussion

In the current research, we examined the effects of a gender bias intervention in which people were confronted with disconcerting evidence about this ongoing problem in society. We consider this intervention an example of how results of scientific research are typically communicated, in order to influence people's attitudes and behaviours: It is common practice to present evidence of the occurrence of the problem—and to emphasize how the self is implied in the persistence of these problems—as a way to convey the relevance and urgency of particular issues. Nevertheless, we hypothesized that confronting people in this way is also likely to raise a threat response which gives rise to defensiveness, denial, avoidance, or passivity. In addition to examining the occurrence of threat, we therefore also examined whether and how people can be helped to overcome such initial stress responses, framing the intended outcome of the intervention differently to increase its impact. In this way, we aimed to gain a better understanding of the psychological mechanisms that are activated when people are confronted with evidence of ongoing problems in society–in this case the existence of gender bias in teaching evaluations. To achieve this aim, we first tested the prediction that people feel threatened when

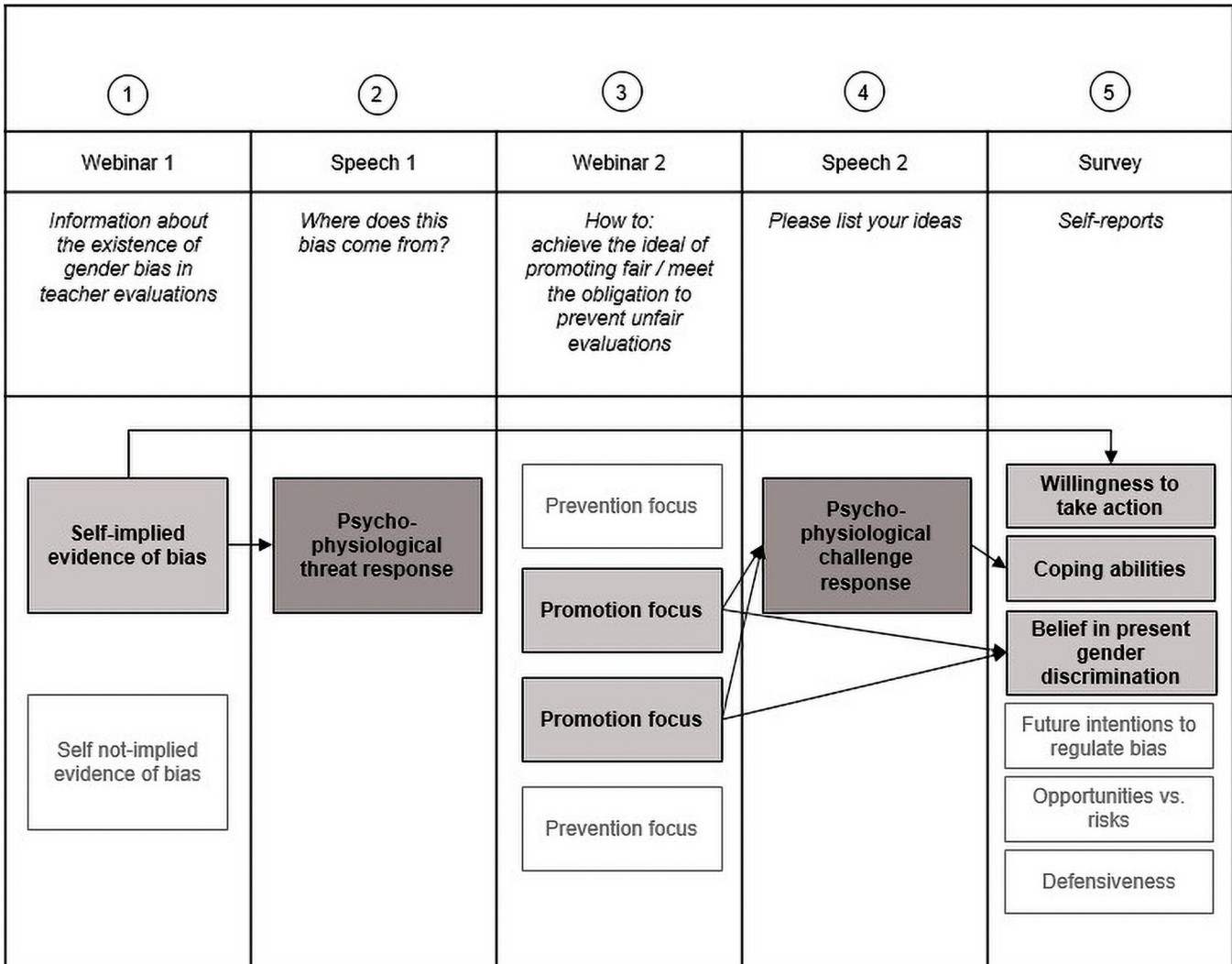

**Fig 3. Overview of the findings.**

presented with evidence of their own group's bias (by means of a *self-implication manipulation*, in the first part of the intervention). Second, we tested whether a promotion focus (vs. prevention focus; *framing manipulation*, in the second part of the intervention) to address the issue of gender bias, would help to elicit a motivational state of challenge. A visualization of the different parts of the intervention and our main findings are displayed in Fig 3.

We found evidence in line with predictions. First, participants who were confronted with self-implying evidence of gender bias (as compared to participants who saw evidence not implying the self) showed an increased initial psychophysiological threat response—as indicated by higher Total Peripheral Resistance (TPR) and relatively lower Cardiac Output (CO). This extends prior research revealing the threat of being reminded of the moral shortcomings of oneself or one's group–which is greater than the threat of being reminded of past competence failures [44]. Further, previous research has shown that people feel particularly threatened and show defensive responses when confronted with information about the moral inadequacy of another ingroup member, compared to when they are confronted with moral failures of outgroup members [34]. In general, when the occurrence of threat and defensive

responses is observed, people become less motivated to act and engage in initiatives to address the topic [36, 38]. However, our study is the first to show how common intervention strategies–in which research evidence of ongoing problems is presented as a way to motivate people towards change–can raise a threat response that can have counterproductive effects.

The second main finding of our study is therefore of utmost importance. In line with our hypothesis, results on the Threat-Challenge Index (TCI) showed how the initial threat response can be overcome. That is, we observed a reliable difference in the cardiovascular responses of participants in the promotion condition and participants in the prevention conditions. Thus, once participants were asked to verbally reflect on ways to *pursue the ideal of promoting* fair evaluations of teachers, they showed relatively more challenge than those who were asked to address the issue by listing potential efforts to *meet the obligation of preventing* unfair evaluations—who showed relatively more threat during this task. The promotion condition also made them more likely to acknowledge that gender bias in teaching evaluations continues to be a problem. This observation extends previous research in which students were asked to reflect on evidence of ongoing labour discrimination, about which their responses differed depending on whether the moral implications of cultural diversity were framed in terms of ideals or obligations [39]. In line with that work, we demonstrate that encouraging a focus on promoting positive outcomes and ideals can make people more receptive of information documenting one's own role and responsibility in the persistence of social problems [34, 39, 40, 42]. This suggests that the counterproductive effects of presenting scientific results to communicate the severity, urgency and self-relevance of a societal problem can be overcome. That is, explicitly adopting a focus on the promotion of ideal and desired outcomes (rather than focusing on the obligation to prevent problems) highlights future opportunities for change. This helps people to acknowledge the severity of ongoing problems and stimulates them to consider and develop strategies for (behavioural) improvement (see also [34, 39, 40])).

Related to the third aim and hypothesis of our research, we examined whether the state of challenge was related to self-reported coping abilities. Our findings showed that–only in the promotion condition–participants who displayed a stronger cardiovascular challenge response also reported feeling more able to cope with the situation. This extends prior work which documented that helping people to focus on future improvements, rather than past failures—as well as presenting their past failures as indicating a lack of competence rather than stemming from a lack of moral intentions [46]—enhances their perceived coping abilities and reduces defensive responses. We observed a similar effect when we invited participants—after completion of the whole intervention—to consider what they might do differently in the future. Here we found that those who had received self-implying evidence of gender bias in teaching evaluations (but regardless of whether they were presented with a promotion or prevention frame to deal with the issue in the future) were more willing to take action, for instance by becoming involved in a student panel to talk about the future of teaching evaluations. We think this extends prior work and shows how important it is for interventions to both highlight the severity and urgency of particular problems—for instance by emphasizing how the self is implied in the persistence of these problems—, as well as to help people look forward to possible solutions and opportunities to change their future behaviours. That is, our findings indicate that in theory, research evidence of ongoing problems, as well as people's own role in perpetuating these problems, can communicate the urgency, severity and relevance of such problems, and enhance their willingness to take action. However, our findings also show the dangers of this approach–when documenting gender bias. That is, examining cardiovascular response patterns at different stages of the intervention, allowed us to reveal an initial threat response when being confronted with self-implying evidence of bias–which in practice can raise defensive, avoidant, and passive responses [36, 37, 38]. Thus, we emphasize the importance of the second

part of our intervention, containing our second experimental manipulation. Here we demonstrated that explicitly emphasizing the promotion of desired outcomes—in this case the ideal of fair evaluations—caused participants to display a physiological challenge response, which was associated with greater self-perceived coping abilities. Furthermore, only in this condition were participants inclined to accept and embrace the evidence shown to them, and acknowledge the perseverance of gender bias as an ongoing problem.

## Theoretical and practical implications

To optimize the impact of interventions that rely on the communication of research findings and provide participants with self-implying evidence of the recurring problem, it is important to examine when and why such interventions may and may not work, and how they can be made more effective. The findings from the current research contribute to these insights by showing how specific aspects of an anti-bias intervention impacts on the experiences of people taking part in the intervention. Thus far, most of the anti-bias interventions that are offered are not systematically evaluated for their impact [6]. And if evaluations are made, these usually consist of proximal subjective ratings, asking individual participants to indicate retrospectively whether they liked taking part in the intervention [79, 80]. Yet from other types of interventions (e.g., aiming to improve people's lifestyle or health behaviours), we know that subjective likeability of the intervention does not necessarily predict the probability of behavioural improvement over time [16, 19, 81]. Studies that do address more distal and objective indicators tend to rely on macro-level archival data (e.g., observed changes over time in organizational level hiring percentages of groups targeted in the intervention [82]. In studies such as these, little information is available about the underlying process of change or the psychological mechanisms through which specific interventions result in these outcomes. The current study thus complements prior efforts by directly examining the process of change participants undergo at different stages of an the intervention. Further, in addition to measuring participants' explicit retrospective evaluations of the intervention by means of self-report questionnaires, we assessed their more implicit responses to different parts of the intervention in *real-time*. That is, we measured participants' physiological states *while* they reflected on the confrontation with (self-implying) evidence of bias, and while they considered ways to either prevent such bias from occurring or to promote unbiased evaluations.

Importantly, our real-time physiological measures revealed a threat response caused by the confrontation with self-implying evidence of bias, as well as a challenge response when considering how to promote more fair evaluations. Importantly, this information on how participants were affected by the intervention could not be captured from the more traditional and deliberate self-report measures. Yet, we did observe the impact of the intervention on participants' self-stated willingness to acknowledge the perseverance of the problem, and their willingness to address it. Thus, this demonstrates the added value of including implicit measures of psychophysiological responses, and capturing changes in the associated motivational states, in response to different aspects of the intervention.

The self-report measures in our study revealed that participants who were confronted with evidence of gender bias in their own and their fellow group members' teaching evaluations, thought this evidence was based on fact and not misleading or exaggerated. However, these participants also reported not to believe that this self-implying evidence of gender bias represented their true values or reflected their true thoughts or feelings. These findings are in line with previous research indicating that bias is often implicit [83, 84], which makes it difficult to confront people with such bias in explicit ways. Yet, many anti-bias interventions are focused on making people aware of such biases to elicit more positive attitudes and behavioural

changes towards disadvantaged groups [14, 15]. Importantly, our psychophysiological findings indicated that this self-implying evidence was associated with a motivational state of threat. Other research has revealed that this may result in less motivation to act due to different interpretations of the evidence of (one's own, or one's group) social bias, depending on people's perspective and ideology and how they view the source of the message. In other words, just giving the facts is not enough to get people to change [18, 85]. The results of the current study therefore not only show the potential downsides of procedures that try to engage people who attend an intervention by confronting them with evidence of (their own) bias, they also reveal that alleviating such threat further helps people to consider possibilities to address the issue.

In terms of practical implications, the findings of the current study are relevant to the notion that 'best practices' regarding diversity and inclusion initiatives are not characterized only by *what*, for instance, organizations do to combat (implicit) bias but also depends on *how* they do this. That is, even though many anti-bias interventions are based on raising awareness about people's own (implicit) biases, our findings show that being confronted with this kind of evidence can trigger threat responses. Moreover, we have shown that reflecting on what ideals to pursue to address the issue of social bias is associated with a motivational state of challenge and greater perceived coping abilities. In other words, anti-bias interventions may include a direct way of making people aware of the presence of bias in their own group, but such evidence is preferably accompanied by the opportunity to address the issue in terms of one's (shared) ideals. For instance, by reflecting on ways to strive for the promotion of fair judgements, rather than in terms of what one is obliged to do by reflecting on one's obligation to prevent bias.

## Limitations and suggestions for future research

Our findings, based on both explicit self-reports as well as more implicit real-time psychophysiological indices of motivational states, underline the importance and added value of having a multi-level approach to examine the responses of participants to anti-bias interventions. That is, consistent with the design of prior research, our self-report measures were assessed after the entire intervention—including the presentation of self-implied vs. self not-implied evidence of bias, and the promotion vs. prevention frame for addressing the bias. They thereby indicate how people reflect on the issue of the existence of gender bias, and how to address it, in relative retrospect. In contrast, our psychophysiological measures are indicative of people's real-time implicit responses, which provided our findings of motivational states of threat and challenge *during* the intervention—which at times diverged from our self-report findings. Then again, we may have found different results if we had included part of the self-reports after the first experimental manipulation in which participants had only been confronted with their own (group's) bias, without having had the opportunity (yet) to think about how to deal with the issue. A suggestion for future research would be to examine the effects of confronting people with their own group's bias, by also including self-report measures of acceptance of that evidence and potential defensiveness towards that evidence, directly after the evidence is presented. This would allow for a more thorough investigation of the extent to which becoming aware of the presence of bias in one's own group is related to participants' automatic psychophysiological threat response as well as their subsequent deliberate reflection on the issue. Relatedly, decided not to include a post-measure of participants' teaching evaluations to test the effectiveness of our intervention, as has been done in prior work on gender bias in teaching evaluations [86, 87]. This was outside the scope of the current research, as we intentionally focused on investigating the underlying psychological mechanisms that may contribute to the varying outcomes of such interventions (as well as the mixed results of anti-bias interventions

in various other contexts [e.g., 14, 15, 88]. Further, we note that the impact of our intervention could be broader than that (e.g., by motivating students to take action against survey methods to evaluate teaching quality due to the realization that these are subject to bias). Nevertheless, future work could include a post-measurement of bias to more specifically assess whether and how these types of awareness raising interventions help to eliminate bias.

Additionally, our current examination addressed a single-session intervention. Future research might explore how this works in a more elaborate approach, where multiple sessions are used to address specific phases and aspects of the intervention, and each are evaluated separately.

Furthermore, future work could extend intervention materials and session characteristics beyond the webinar paradigm we used in the current study. That is, we asked participants to individually view a webinar about gender bias in teacher evaluations and to verbally reflect on this gender bias via a computer screen. An alternative way of presenting such an intervention is during an actual group event, allowing for real time interactions or workgroup sessions with other attendees. This is another option to explore in future research, where multiple sessions can be used to compare the impact of different types of interventions (see also [89]). Nevertheless, considering that many of such events have been replaced by webinars or other online meetings since the Covid-19 pandemic, our experimental paradigm nicely reveals how such a 'distant' approach to anti-bias interventions can still have impact and be effective. In fact, we think this approach can be extended to other topics and issues where research findings are used as an intervention to convey the severity and urgency of ongoing social problems. Indeed, the results of our investigation indicate that the social impact and benefits of research findings can be enhanced when paying more attention to the psychological effects of science communications on its recipients.

## Conclusion

In conclusion, our research shows that anti-bias interventions that include self-implying evidence of the presence of bias may enhance the perceived urgency and relevance of the problem that might benefit participants' willingness to take action. However, we also demonstrated that this is likely to elicit a threat response, which is associated with denial, avoidance, and other maladaptive responses. Importantly therefore, this work elucidates the added value of presenting research evidence of bias *in combination with* the invitation to think about ways to *promote ideal outcomes*. Indeed, an intervention that combines these two aspects is most likely to result in a motivational state of challenge, where people are willing to acknowledge that gender discrimination is an ongoing problem, but also experience the ability to cope with this problem by taking action.

## Supporting information

**S1 Appendix. Teacher evaluation survey.**
(DOCX)

**S2 Appendix. Additional measures lab session.**
(DOCX)

**S3 Appendix. Speech data.**
(DOCX)

## Acknowledgments

We want to thank the following (former) members of the Organizational Behavior group for their help with data collection, data preparation, and/or preliminary data analyses (in

alphabetical order): Jamie Breukel, Nadia Buiter, Martine Kloet, Wiebren Jansen, Inga Rösler, Jeanette van der Lee, and Marleen van Stokkum. In addition, we thank Ronny Ramos Delgado for his help with the preparation of the manuscript for submission. Furthermore, we want to thank other (former) members of the Organizational Behavior Group with whom we discussed the idea, conceptualization, methodology and write-up of this research (in alphabetical order): Tatiana Chopova, Tessa Coffeng, Florien Cramwinckel, Piet Groot, Onur Sahin, Elianne van Steenbergen, and Melissa Vink. The current Organizational Behavior Group membership list can be found here: https://www.uu.nl/en/research/organisational-behaviour/our-team. Lead author for this group is Naomi Ellemers (contact: n.ellemers@uu.nl).

## Author Contributions

**Conceptualization:** Félice van Nunspeet, Esmee M. Veenstra, Naomi Ellemers, Daan Scheepers, Miriam I. Wickham, Jojanneke van der Toorn.

**Data curation:** Esmee M. Veenstra, Beatriz Monteiro Graça Casquinho.

**Formal analysis:** Félice van Nunspeet, Esmee M. Veenstra, Beatriz Monteiro Graça Casquinho, Daan Scheepers, Miriam I. Wickham, Elena A. M. Bacchini, Jojanneke van der Toorn.

**Funding acquisition:** Naomi Ellemers.

**Investigation:** Esmee M. Veenstra, Miriam I. Wickham.

**Methodology:** Félice van Nunspeet, Esmee M. Veenstra, Naomi Ellemers, Daan Scheepers, Miriam I. Wickham, Jojanneke van der Toorn.

**Project administration:** Esmee M. Veenstra.

**Supervision:** Félice van Nunspeet, Naomi Ellemers.

**Validation:** Félice van Nunspeet, Esmee M. Veenstra, Beatriz Monteiro Graça Casquinho, Miriam I. Wickham, Elena A. M. Bacchini.

**Visualization:** Félice van Nunspeet, Esmee M. Veenstra.

**Writing – original draft:** Félice van Nunspeet, Esmee M. Veenstra, Beatriz Monteiro Graça Casquinho, Naomi Ellemers, Daan Scheepers, Miriam I. Wickham, Elena A. M. Bacchini, Jojanneke van der Toorn.

**Writing – review & editing:** Félice van Nunspeet, Esmee M. Veenstra, Beatriz Monteiro Graça Casquinho, Naomi Ellemers, Daan Scheepers, Miriam I. Wickham, Elena A. M. Bacchini, Jojanneke van der Toorn.

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
