## [Decision Letter · Decision Letter 0]

14 Aug 2024

PONE-D-24-01387Overcoming the threat of anti-bias interventions:

Combining self-report and psychophysiological measures to capture the process of change.

PLOS ONE

Dear Dr. van Nunspeet,

Thank you for submitting your manuscript to PLOS ONE. After careful consideration, we feel that it has merit but does not fully meet PLOS ONE’s publication criteria as it currently stands. Therefore, we invite you to submit a revised version of the manuscript that addresses the points raised during the review process.

My apologies for the delay in reaching a decision. I have carefully read the manuscript, along with the review we received, and I found both to be highly insightful. I agree with the reviewer’s constructive feedback and encourage the authors to address their concerns thoroughly.

In my assessment, I found the study to be scientifically sound and comprehensive. While we do not evaluate submissions based on their significance alone, I would like to highlight that this paper is both innovative and important. It challenges the conventional assumptions around anti-bias training by offering a fresh perspective, emphasizing the significance of the delivery method and the psycho-physiological reactions of participants. Without careful attention to these aspects, such training could result in underwhelming or even counterproductive outcomes.

I am pleased that the authors have chosen PLOS ONE as the venue for this valuable work.

We look forward to receiving your revised manuscript.

Kind regards,

katsuya oi, PhD

Academic Editor

PLOS ONE

3. One of the noted authors is a group or consortium [The Organizational Behaviour Group]. In addition to naming the author group, please list the individual authors and affiliations within this group in the acknowledgments section of your manuscript. Please also indicate clearly a lead author for this group along with a contact email address.

Additional Editor Comments:

My apologies for the delay in reaching a decision. I have carefully read the manuscript, along with the review we received. I agree with the reviewer’s constructive feedback and encourage the authors to address their concerns thoroughly.

In my own assessment, I found the study to be scientifically sound and comprehensive. While we do not evaluate submissions based on their significance alone, I would like to highlight that this paper is both innovative and important. It challenges the conventional assumptions around anti-bias training by offering a fresh perspective, emphasizing the significance of the delivery method and the psycho-physiological reactions of participants. Without careful attention to these aspects, such training could result in underwhelming or even counterproductive outcomes.

I am pleased that the authors have chosen PLOS ONE as the venue for this valuable work.

Reviewers' comments:

Reviewer's Responses to Questions

**Comments to the Author**

1. Is the manuscript technically sound, and do the data support the conclusions?

Reviewer #1: Yes

2. Has the statistical analysis been performed appropriately and rigorously? 

Reviewer #1: Yes

3. Have the authors made all data underlying the findings in their manuscript fully available?

Reviewer #1: Yes

4. Is the manuscript presented in an intelligible fashion and written in standard English?

Reviewer #1: Yes

5. Review Comments to the Author

Reviewer #1: This manuscript attempt to test for physiological responses to different interventions to minimize implicit biases. It uses an example of biases in evaluations of teaching and uses a 2x2 design to test how the implication that the respondents were biased and the framing of solutions changed these responses. The authors find some evidence that the type of intervention does seem to matter with the self implied evidence of bias increasing a threat response while a promotion focused frame increases the challenge response.

This is a technically sound manuscript. The experiment is well done and the results are pretty clear. I would like to see a little more information about the evaluations that the respondents were accused of being biased about. Were they connected to the professor running the experiment? I can imagine that a respondent who was told that they were biased in their evaluations of a professor by that professor may see this as a threat in ways that they would not have if it was by a more neutral arbiter. I am guessing that the researchers did create some space between the person running the experiment and the target of the evaluations, about it should be clear.

The choice of teaching evaluations as the target here is dicey. There is some evidence that interventions to reduce implicit biases can improve the scores for female faculty (Peterson et al 2019 “Mitigating gender bias in student evaluations of teaching” PLoS ONE), but a different widespread test found no effect of these interventions (Genetin et al 2021 “Mitigating implicit bias in student evaluations: A randomized intervention” Applied Economic Perspectives and Policy). Does the effectiveness of the intervention matter for the results? The authors have some self reports, but both of those other papers have experimental evidence about the effectiveness on actual evaluations. If, for instance, the Generin paper is correct and the interventions do not actually work, then some kind of negative reaction seems appropriate from the respondents. In contrast, if the intervention would have actually reduced the implicit biases, the effects may have been different.

6. PLOS authors have the option to publish the peer review history of their article (what does this mean?). If published, this will include your full peer review and any attached files.

Reviewer #1: No

---

## [Author Response · Author response to Decision Letter 0]

30 Oct 2024

Dear Dr. Katsuya Oi, dear Academic Editor of PLOS ONE,

Enclosed we submit the revised version of our manuscript (Manuscript ID PONE-D-24-01387 “Overcoming the threat of anti-bias interventions: Combining self-report and psychophysiological measures to capture the process of change.”) to be considered for publication in PLOS ONE. 

We want to thank you for handling our submission to PLOS ONE, and for the opportunity to revise and resubmit our manuscript. We are very thankful for the positive evaluation and constructive comments. We have noted each comment in response to our initial manuscript, and at your invitation to prepare a revision, we have addressed all the issues that were raised. We feel that we were able to successfully deal with the outstanding questions and requirements. Below, we indicate how we addressed these points one by one. 

Reviewer’s comments:

A) The Reviewer indicated they wanted “to see a little more information about the evaluations that the respondents were accused of being biased about.” The Reviewer wondered whether participants were connected to the professor who run the experiment, as it could be expected that “a respondent who was told that they were biased in their evaluations of a professor by that professor may see this as a threat in ways that they would not have if it was by a more neutral arbiter.” Consistent with the Reviewer “guessing that the researchers did create some space between the person running the experiment and the target of the evaluations”, we ensured this in the experiment. Moreover, in line with the Reviewer’s remark that we should be clearer about this, we included the following information in the method section of our revised manuscript, on page 12: 

“Importantly, none of the researchers (i.e., the experimenters and the university professor who presented in the webinar), was included in participants’ teaching evaluations as these evaluations concerned courses outside of the researchers’ teaching responsibilities. Independence between the researchers and the research participants was thus assured.”

Additionally, we expanded our explanation of how the information about the self-implying evidence was presented (i.e., not in the webinar, or by the professor). On page 12, we added the last sentence below:

“Participants in the self-implied condition were also presented with a brief summary indicating they and their fellow students had shown a gender bias in their teacher evaluations – allegedly based on the online questionnaire they had filled out before they had come to the lab. This feedback was not presented within the webinar (i.e., by the professor), but rather provided as a separate (additional) section of the experiment.” 

B) The second, and last, comment of the Reviewer concerns the fact that whereas we focused on teaching evaluations as the target of our gender bias intervention, we did not test whether our intervention affected such teaching evaluations. Furthermore, the Reviewer poses that the potential results of such a test would have implications for the interpretations of our findings. More specifically, the Reviewer mentions that “there is some evidence that interventions to reduce implicit biases can improve the scores for female faculty (Peterson et al 2019 “Mitigating gender bias in student evaluations of teaching” PLoS ONE), but a different widespread test found no effect of these interventions (Genetin et al 2021 “Mitigating implicit bias in student evaluations: A randomized intervention” Applied Economic Perspectives and Policy).” The Reviewer then wonders whether “the effectiveness of the intervention matter for the results?” 

We thank the Reviewer for alerting us to this matter and for the references to the different findings regarding the effectiveness of gender bias interventions in teaching evaluations. Although we consider these questions as highly relevant in a broader sense, with regard to the more concrete aims of the current contribution, we note that we intentionally focused on a somewhat different question, namely the processes underlying the potential (in)effectiveness of such interventions, as we describe on pages 2 and 3 of the original (as well as the revised) manuscript (“More specifically, we measure how psychophysiological responses unfold during the intervention to capture the process of change, and we combine this with self-reports revealing how participants explicitly reflect on their experience. This approach increases our understanding of psychological mechanisms that may either undermine or enhance the likelihood that people engage with the intervention and adapt their behaviour.”). In other words, the aim of our study was not to test the effectiveness of (the parts of) the intervention that was presented in our experimental design (i.e., the confrontation with self-implying vs. not self-implying evidence of bias, and the reflection on potential solutions in terms of ideals vs. obligations) on the elimination of biased judgments in subsequent course evaluations by these students. Instead, our measure of ‘willingness to take action’ to capture participants’ intentions towards reduction of bias in teaching assessments was framed more broadly. Notably, such action might also include other types of initiatives, such as taking action to change current methods to evaluate teaching quality, due to the realization that these are subject to bias. For future research, we see the potential value of additional measures to examine the impact of our intervention more specifically. We have therefore included this reflection in the revised version of our manuscript (including the references mentioned by the Reviewer), in the paragraph on future research directions. Specifically, on page 34, we now write:

“Relatedly, decided not to include a post-measure of participants’ teaching evaluations to test the effectiveness of our intervention, as has been done in prior work on gender bias in teaching evaluations [86, 87]. This was outside the scope of the current research, as we intentionally focused on investigating the underlying psychological mechanisms that may contribute to the varying outcomes of such interventions (as well as the mixed results of anti-bias interventions in various other contexts [e.g., 14, 15, 88]. Further, we note that the impact of our intervention could be broader than that (e.g., by motivating students to take action against survey methods to evaluate teaching quality due to the realization that these are subject to bias). Nevertheless, future work could include a post-measurement of bias to more specifically assess whether and how these types of awareness raising interventions help to eliminate bias.” 

In addition to the comments made by the Reviewer, we also addressed the additional requirements you listed to prepare our manuscript for publication in PLOS ONE. We detail the changes that were made below. 

1) We carefully checked the PLOS ONE style templates and have ensured that our manuscript meets PLOS ONE's style requirements, including those for file naming. We also made sure to refer to these files using the correct names in the revised manuscript. Please note that this also implies that we have newly uploaded our Figures and Supporting Information documents, to adhere to the correct resolution requirements and file naming respectively. In terms of content, these files did not change. 

2) Consistent with our prior indication that we will make our data available on acceptance, we still aim to adhere to your open data policy. We have prepared a folder to be uploaded in OSF upon publication, that contains 1) the dataset and syntax for all the analyses reported in the main manuscript, and 2) the supporting information documents as well as the syntax for the analyses reported here (and which can be applied to the dataset mentioned under 1).

3) In line with your request about our research group as one of the authors, as well as information in the formatting requirements (i.e., "If there is a consortium or group author on your manuscript, please provide a note that describes where the full membership list is available for the readers."), we have listed all the contributors to the manuscript from The Organizational Behavior Group in the acknowledgments section of our manuscript. In addition, we added a note that specifies where the full membership list is available for the readers, and we have added the lead author for this group along with a contact email address. This can be found in the Acknowledgments section on page 36 of the revised manuscript. We hope these changes meet your request, but please inform us in case additional adjustments need to be made with regard to this issue.

4) We have, once again, reviewed our reference list to ensure that it is complete and correct. Compared to the initial manuscript, we did add three references in our revision, but we have made sure to correctly renumber all references as well as the corresponding reference list. Finally, we noticed that we initially included a URL when mentioning a DOI. Consistent with the criteria and guidelines for formatting references, we now provide DOI’s while adhering to the format with both the label and full DOI included at the end of the reference (and not providing the URL). 

Once again, we want to thank you and the Reviewer for the positive and constructive comments. We did our best to incorporate the suggestions and requests in the revised manuscript and we feel that the suggestions have further improved the quality of our paper. We look forward to your reply.

Sincerely,

also on behalf of Esmee Veenstra (shared first author), 

and our co-authors: Beatriz Casquinho, Naomi Ellemers, Daan Scheepers, Miriam Wickham, Elena Bacchini, and Jojanneke van der Toorn,

Félice van Nunspeet

Corresponding author

---

## [Decision Letter · Decision Letter 1]

18 Nov 2024

Overcoming the threat of anti-bias interventions:

Combining self-report and psychophysiological measures to capture the process of change.

PONE-D-24-01387R1

Dear Dr. van Nunspeet,

We’re pleased to inform you that your manuscript has been judged scientifically suitable for publication and will be formally accepted for publication once it meets all outstanding technical requirements.

Kind regards,

katsuya oi, PhD

Academic Editor

PLOS ONE

Additional Editor Comments (optional):

Reviewers' comments:

Reviewer's Responses to Questions

**Comments to the Author**

1. If the authors have adequately addressed your comments raised in a previous round of review and you feel that this manuscript is now acceptable for publication, you may indicate that here to bypass the “Comments to the Author” section, enter your conflict of interest statement in the “Confidential to Editor” section, and submit your "Accept" recommendation.

Reviewer #1: All comments have been addressed

2. Is the manuscript technically sound, and do the data support the conclusions?

Reviewer #1: Yes

3. Has the statistical analysis been performed appropriately and rigorously? 

Reviewer #1: Yes

4. Have the authors made all data underlying the findings in their manuscript fully available?

Reviewer #1: Yes

5. Is the manuscript presented in an intelligible fashion and written in standard English?

Reviewer #1: Yes

6. Review Comments to the Author

Reviewer #1: The authors have addressed all of my comments and I think the paper should be accepted for publication.

7. PLOS authors have the option to publish the peer review history of their article (what does this mean?). If published, this will include your full peer review and any attached files.

Reviewer #1: No

---

## [Editor Report · Acceptance letter]

2 Jan 2025

PONE-D-24-01387R1 

PLOS ONE

Dear Dr. van Nunspeet, 

I'm pleased to inform you that your manuscript has been deemed suitable for publication in PLOS ONE. Congratulations! Your manuscript is now being handed over to our production team.

Kind regards, 

on behalf of

Dr. katsuya oi 

Academic Editor

PLOS ONE